# Cross-Cultural Adaptation and Psychometric Testing of the International Sedentary Assessment Tool for the Spanish Population

**DOI:** 10.3390/ijerph17030758

**Published:** 2020-01-24

**Authors:** Antonio I. Cuesta-Vargas, Cristina Roldán-Jiménez, Jaime Martín-Martín, Manuel González-Sánchez, Daniel Gutiérrez Sánchez

**Affiliations:** 1Department of Physiotherapy, University of Málaga, 29071 Málaga, Spain; cristina.roldan005@gmail.com (C.R.-J.); mgsa23@uma.es (M.G.-S.); 2Biomedical Research Institute of Málaga (IBIMA), 29071 Málaga, Spain; jaimemartinmartin@gmail.com (J.M.-M.); danieltunie@hotmail.com (D.G.S.); 3Institute of Health & Biomedical Innovation, Queensland University of Technology, Queensland 4059, Australia; 4Department of Legal Medicine and Anatomy, University of Málaga, 29071 Málaga, Spain; 5Department of Nursing and Podiatry, University of Málaga, 29071 Málaga, Spain;

**Keywords:** sedentary behavior, physical activity, outcome measures, psychometric properties, validity, questionnaire

## Abstract

Sedentary behavior (SB) is currently considered a public health problem with a high cost of care. Evaluating SB is essential for prevention and early management of physical inactivity. The International Sedentary Assessment Tool (ISAT) is an instrument that has been developed to assess SB. The aim of this study was to carry out a cross-cultural adaptation and a psychometric analysis of the Spanish version of the ISAT. A cross-sectional study was conducted. A total of 432 participants were included in this study. A double forward method and a backward method were used to translate the ISAT. A psychometric analysis of internal consistency and concurrent criterion validity was performed according to the most up-to-date Consensus-based Standards for the Selection of Health Measurement Instruments (COSMIN). No language difficulties were found in the translation process. The Spanish version of ISAT was readable and acceptable. Internal consistency was satisfactory (α = 0.80). Criterion validity was demonstrated (rho=0.63). The Spanish version of the ISAT is a valid and reliable measure that can be used clinically to assess SB. Further studies assessing other psychometric properties are needed.

## 1. Introduction

Sedentary behavior (SB) is currently considered a public health problem with a high cost of care, which is related to technological advances and industrialization [1].

SB includes activities performed while awake that involve sitting, reclining, and lying down, and which require low energy consumption [2]. Despite the health effects associated with regular physical exercise, SB remains a common problem [3,4]. In this context, SB behavior is an important risk factor for several chronic diseases and mortality, which explains the interest in studying the effect of SB in both the development and progression of risk factors in chronic diseases that have increased in recent years [5,6,7]. Currently, non-communicable diseases (NCDs), such as cardiovascular disease, cancer, respiratory diseases, and diabetes lead most causes of death and disability worldwide [8,9,10]. Such diseases account for the vast majority of the burden of disease and premature mortality [11]. Both issues can be partially countered by behavioral risk factors, with physical inactivity among them [12]. It is therefore vital for clinicians to measure SB in order to establish the effectiveness of behavioral change interventions among the population [13,14].

The use of questionnaires has been one of the most frequent tools to measure and quantify SB [15,16]. Some SB questionnaires have been developed to assess different dimensions of this construct, such as time invested during leisure (watching television and playing video games or using computer screens), and time spent sitting during the workday [15,17,18]. In this context, the International Sedentary Assessment Tool (ISAT) is an instrument that has been recently proposed to assess SB on population health surveys [15]. The ISAT is a questionnaire formed by the most valid and reliable questions related to specific aspects of SB identified after a systematic review that included 49 questionnaires, aimed at evaluating SB in both adults (35 questionnaires) and pediatric (14 questionnaires) patients [15]. However, the psychometric properties of the ISAT itself have not been assessed. One of the main barriers to measuring SB in Spain has been to find valid, reliable instruments that can capture the complexity of this construct. As a result, there is limited information available on assessing SB in Spanish culture. The aim of the study was therefore to carry out a cross-cultural adaptation of the ISAT into Spanish (ISAT-Sp) as well as a psychometric analysis of the Spanish population.

## 2. Materials and Methods

### 2.1. Study Design and Participants

A cross-sectional design was used to conduct this study, which consists of two main phases: (1) translation and cross-cultural adaptation and (2) psychometric testing.

The inclusion criteria were: (1) Spanish-speaking adults (age ≥ 18 years), and (2) people who had signed an informed consent. The exclusion criteria were: (1) the participant’s refusal to take part in the study, and (2) the participant being under 18 years old.

Participants were recruited from different social environments in Malaga (Spain). University students, university professors, administration, and services staff and clinical professionals from different health centers participated in this study. Potential participants received an online survey to fill out with the ISAT and the sociodemographic information. Data were obtained between May and June 2019.

Standards of good clinical practice and the ethical principles established for research on human beings were maintained at all times.

### 2.2. Translation and Cross-Cultural Adaptation

The translation was carried out in accordance with the forward and backward method [19]. Firstly, two independent native Spanish speakers translated the ISAT from English to Spanish. The two forward versions were compared and, after a consensus, a preliminary Spanish version of the ISAT-Sp was created. The back translations (Spanish into English) were done independently by two native English translators. Minor discrepancies were discussed and resolved. The prefinal version of the ISAT-Sp was pilot-tested to ensure the Spanish version of the ISAT was comprehensible and acceptable (Figure 1).

### 2.3. Psychometric Testing

Four hundred and thirty-two participants were included in this phase. A psychometric analysis of internal consistency and criterion validity was performed, according to the Consensus-based Standards for the Selection of Health Measurement Instruments (COSMIN) [20]. The following hypotheses were tested.

The overall internal consistency of the questionnaire will be 0.70 or above, and the intra-class correlation coefficient will be 0.70 or above.The ISAT will be positively correlated with the time that participants have spent sitting on a weekday during the last seven days, with a minimum value of a correlation coefficient of 0.50.

### 2.4. International Sedentary Assessment Tool

The ISAT is an instrument that has been developed to assess SB [15]. This instrument has been recommended for use in population health surveys based on the best available evidence. The ISAT consists of two parts. The first part comprises six items. The participant is asked about his or her activities on a typical weekday in the last week while sitting, reclining, or lying down. The second part also comprises six items. The participant is asked about his or her activities on a typical weekend day in the last week while sitting, reclining, or lying down. Higher scores involve more SB [15].

### 2.5. Data Analysis

A descriptive analysis of demographic and clinical variables was performed. We determined the distribution and normality of the sample by performing a one-sample Kolmogorov–Smirnov (KS) test. Cronbach’s α coefficients and Intraclass Correlation Coefficient Type 2.1 (ICC_2.1_) were calculated to determine the internal consistency of ISAT [21].

Criterion validity was determined using the bivariate (Spearman) correlation coefficient. The correlation coefficient used the criteria of poor (rho < 0.49), fair (rho = 0.50–0.74), and strong (rho > 0.75) [22]. Data was entered into a data file and analysed using an SPSS statistical program (version 20).

## 3. Results

### 3.1. Participants

The KS test (Asymp. Sig. (2-tailed) = 0.004) indicated that the sample was not normally distributed. A total of 552 surveys were distributed and 452 surveys were returned. Twenty cases were removed due to items missing (more than 25%). The sample population participating in this study comprised 432 adults, aged 35.42 (±10.51) years. Descriptive and anthropometric variables are shown in Table 1.

### 3.2. Translation and Cross-Cultural Adaptation

The ISAT was translated and back-translated without language difficulties to provide the Spanish version of this instrument. Minor discrepancies were discussed and resolved during face validity and feedback. This instrument proved to be comprehensible and acceptable among the target population (Appendix A).

### 3.3. Psychometric Testing

#### 3.3.1. Internal Consistency

The overall internal consistency of the questionnaire was 0.80, and the intra-class correlation coefficient was 0.80 (95% CI 0.75 to 0.84) (Table 2).

#### 3.3.2. Criterion Validity

Concurrent criterion-related validity was determined from the relationship between the ISAT and the time that participants have spent sitting on a weekday during the last seven days. Concurrent criterion-related validity was demonstrated, with a fair and positive correlation of rho = 0.63.

## 4. Discussion

There is limited information available on assessing SB in the Spanish culture, due to the lack of valid, reliable instruments to measure this construct. This study aimed to translate and validate the ISAT into Spanish, and it is the first report involving the translation and validation of this instrument in Spain. In this context, this study provides evidence for the validity of an instrument for assessing SB in Spanish people, which is essential for the prevention and early management of physical inactivity [15].

The psychometric analysis of the ISAT-Sp was carried out satisfactorily, according to the COSMIN standard [20]. In this regard, the ISAT-Sp showed satisfactory psychometric properties in reference to cross-cultural validity, internal consistency [23], and criterion validity [24,25].

The ISAT-Sp has proven to be comprehensible and acceptable. This instrument is simple to complete and easily understood.

The internal consistency was satisfactory (Cronbach’s α = 0.80), and the intra-class correlation coefficient was 0.80 (95% CI 0.75 to 0.84). According to Prince et al., it is ideal to have an ICC and Cronbach’s α as close to 1 as possible, with values over 0.75 considered excellent, for a measure of SB in population health surveys [15]. In this regard, the values of ICC and Cronbach’s α for the ISAT-Sp were excellent and were higher than those reported in other studies on the validation of instruments assessing SB [15,17,26,27].

Criterion validity analysis was supported by a fair correlation (rho=0.63), which provides evidence of construct validity. Validity of a self-report SB measures have been often assessed against an objective measure such as an accelerometer [15]. Validation studies of self-report SB measures have shown poor criterion validity when these measures have been assessed against objective measures [15]. In this context, the values of criterion validity for the ISAT-Sp were high when compared to other studies on the validation of instruments assessing SB [15,28,29].In comparison with other measures, which have been validated for assessing SB in the Spanish population, our results showed higher values of convergent validity than those found by Munguia-Izquierdo et al. [30].

There is a need for the development of valid and reliable instruments for measuring SB in order to provide accurate, consistent measures over time [31]. These instruments must be concise, valid, reliable, evidence-based, and developed using best practices [31]. In this context, the ISAT is a measure that was recently proposed to assess SB on population health surveys [15]. The ISAT was developed based on the most valid and reliable questions for targeting important modes of SB by systematically reviewing the literature to identify measures with the best psychometric properties. This study provides evidence for the validity of the ISAT-Sp. In this regard, the ISAT-Sp is a concise, valid, reliable, evidence-based instrument developed using best practices. Thus, the ISAT-Sp is a measure that can be recommended to assess SB on population health surveys in the Spanish population. Future research should study ISAT in different clinical populations of NCDs, such as cardiovascular disease or cancer, and different musculoskeletal conditions, such as fibromyalgia [32].

### 4.1. Study Strengths and Limitations

To the best of our knowledge, this is the first validation of the ISAT in the Spanish population. Among the strengths of this study are the large sample size (*N* = 432). Moreover, this study was carried out following international recommendations.

There are some limitations on this study. The main limitation is the lack of a device, such as an accelerometer, to objectively measure physical activity. Future studies could be designed to keep an objective record of the time that the patient effectively spends in SB, considering that the inclinometers are able, even without movement, to differentiate between sitting and stationary standing.

This is a validation study, where internal consistency and criterion validity have been assessed. Further studies assessing other psychometric properties in different clinical populations of NCDs are needed.

### 4.2. Implications for Future Research

The current study demonstrates that the ISAT-Sp is a valid measure for assessing SB in the Spanish population, which allows researchers to evaluate this construct. SD research can increase our understanding of this construct, which is essential for the prevention and early management of physical inactivity. In this regard, SD research is an area of special interest since it has implications for the development of interventions to reduce SD in this population. Test-retest reliability, structural validity, and sensitivity to change have not been tested in this study. Further studies assessing these properties should, therefore, be carried out. Furthermore, future research should study the ISAT in different clinical populations of NCDs, such as cardiovascular disease or cancer, and different musculoskeletal conditions, such as fibromyalgia.

## 5. Conclusions

The Spanish version of the ISAT is a valid and reliable measure that can be used clinically to assess SB. This instrument is comprehensive and can capture the multidimensional aspects of a range of symptoms. Further studies assessing other psychometric properties in different clinical populations of NCDs are needed.

## Figures and Tables

**Figure 1 ijerph-17-00758-f001:**
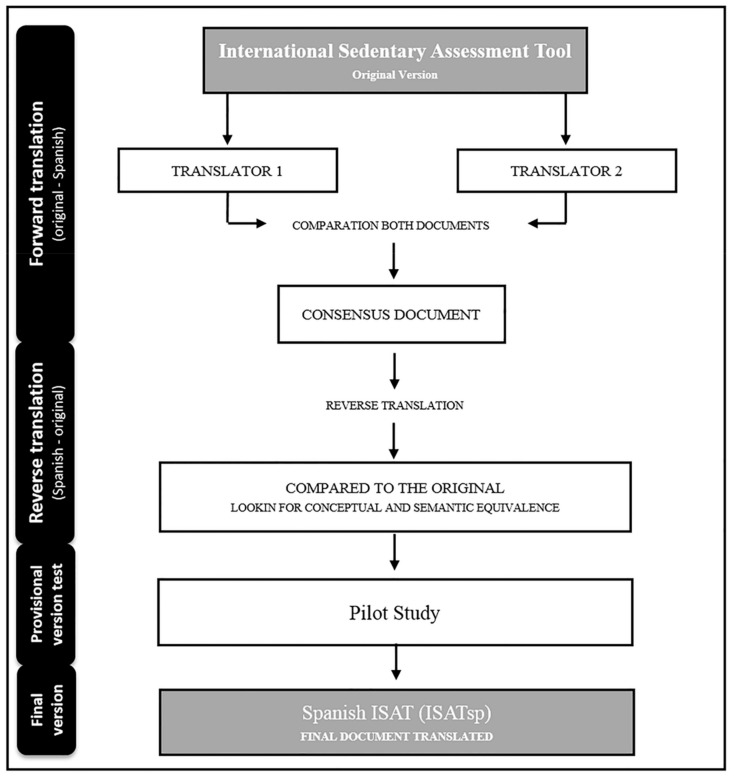
Flowchart of the development process ISAT Spanish version (ISATsp).

**Table 1 ijerph-17-00758-t001:** Characteristics of the participants (*N* = 432).

Variable	Mean SD or Frequency (%)
Age	35.42 (10.51)
Height (M)	1.68 (0.22)
Weight (kg)	70.77 (14.57)
Body mass index (kg/m^2^)	23.89 (4.50)
Sex	
Male	198
Female	228
Profession	
Health professional	293
Office staff	100
Others	39
Health problems	
Asthma	22
Back and neck pain	77
Eye and ear problems	38
Headache	33

Note. SD: Standard deviation.

**Table 2 ijerph-17-00758-t002:** Descriptive statistics and internal consistency for items from the ISAT.

ISAT Items	Mean ± SD	Corrected Item-Total Correlation	Cronbach’s Alpha if Item Deleted
On a typical WEEKDAY in the past week, how much time did you spend sitting or lying down and *En un típico DÍA LABORAL en la semanapasada, ¿cuántotiempo pasó sentado o acostado y*			
1. Watching TV or using a computer, tablet, or smartphone? *Viendo la televisión o usando un ordenador, tablet o smartphone*	2.61 ± 3.39	0.61	0.77
2. Watching television or videos? *Viendo la televisión o vídeos*	2.61 ± 0.38	0.60	0.76
3. Using a computer? *Utilizando el ordenador*	3.69 ± 4.14	0.52	0.77
4. Sitting reading a book or magazine? *Sentado leyendo un libro o revista*	0.96 ± 0.42	0.33	0.79
5. During the last seven days, how much time did you usually spend sitting on a weekday? *Durante los últimos 7 días, ¿cuánto tiempo solía pasar sentado en un día laboral?*	4.65 ± 4.15	0.46	0.78
6. Sitting and driving in a car, bus, or train? *Sentarse y conducir en un coche, autobús o tren*	1.64 ± 2.38	0.37	0.78
On a typical WEEKEND DAY in the past week, how much time did you spend sitting or lying down and En un típico DÍA de FIN DE SEMANA, ¿cuánto tiempo pasó sentado o acostado y			
1. Watching TV or using a computer, tablet, or smartphone? *Viendo la televisión o usando un ordenador, tablet o smartphone*	3.10 ± 2.83	0.51	0.77
2. Watching television or videos? *Viendo la televisión o vídeos*	3.11 ± 2.82	0.50	0.78
3. Using a computer? *Utilizando el ordenador*	2.27 ± 2.17	0.38	0.78
4. Sitting reading a book or magazine? *Sentado leyendo un libro o revista*	1.05 ± 1.18	0.26	0.80
5. During the last seven days, how much time did you usually spend sitting on a weekday? *Durante los últimos 7 días, ¿cuánto tiempo solía pasar sentado en un día laboral?*	4.32 ± 3.29	0.51	0.78
6. Sitting and driving in a car, bus, or train? *Sentarse y conducir en un coche, autobús o tren*	1.22 ± 1.59	0.34	0.79

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
