# Peer review of "Cross-Cultural Adaptation and Psychometric Testing of the International Sedentary Assessment Tool for the Spanish Population"

_ijerph, 2020, doi:10.3390/ijerph17030758_

Round 1
Reviewer 1 Report
I would like to thank the authors for undertaking a study to validate the ISAT for people living in Spain. While the study holds scientific merit, I have a few comments that could improve the information presented in the manuscript.
In the abstract in lines 17 and 18, Sedentary behavior is abbreviated as SD instead of SB
In line 17, Please improve the grammar, "higher care cost" should "higher cost of care
In the methods section (lines 61 - 70), please provide information on how the sample were recruited is missing. Are they a reasonable representation of people in Spain or in a specific region?
Also in the methods section, please provide an example of the English and Spanish versions of the questionnaire.
Also provide an example of what types of questions were used to test internal consistency in both English and Spanish
Finally, can you provide justification on why the COSMIN standards are appropriate for evaluating the internal consistency and criterion validity of the questionnaire. In the citation you provided, the conclusions of the paper pasted below indicates that the COSMIN methods are for systematic reviews and not for the type of work presented in the manuscript.
"The COSMIN Risk of Bias checklist was developed exclusively for use in systematic reviews of PROMs to distinguish this application from other purposes of assessing the methodological quality of studies on measurement properties, such as guidance for designing or reporting a study on the measurement properties."
Author Response
ITEMIZED LIST OF THE REVIEWER COMMENT
International Journal of Environmental Research and Public Health
Title: Cross-cultural adaptation and psychometric testing of theInternational Sedentary Assessment Tool for the Spanish population.
Version: 2
Date: 07thJanuary 2020
Authors: Dr. Antonio Cuesta-Vargas *, Cristina Roldan-Jimenez, JaimeMartin-Martin, Manuel Gonzalez-Sanchez, Daniel Gutiérrez-Sánchez.
Dear Editors,
We would like to thank you and the reviewers for the review of our manuscript, “Cross-cultural adaptation and psychometric testing of the International Sedentary Assessment Tool for the Spanish population” and the opportunity to revise it. As requested, below we provide a point-by-point response to all reviewer comments and questions. We have added a revised manuscript as well as a copy in which the changes are indicated in red. We believe these revisions have improved the manuscript and look forward to your decision. We are happy to make additional revisions if necessary.
Reviewer 1
I would like to thank the authors for undertaking a study to validate the ISAT for people living in Spain. While the study holds scientific merit, I have a few comments that could improve the information presented in the manuscript.
In the abstract in lines 17 and 18, Sedentary behavior is abbreviated as SD instead of SB Authors: We thank you for your According to your comment, we have stated “SB” instead of “SD”.
In line 17, Please improve the grammar, "higher care cost" should "higher cost of care. Authors: We thank you for your Following your suggestion, we have stated “high cost of care” instead of “high care cost”.
In the methods section (lines 61 - 70), please provide information on how the sample were recruited is missing. Are they a reasonable representation of people in Spain or in a specific region? Authors: We thank you for your We have added the following sentence to clarify how the sample were recruited “The participants were recruited from different social environments in Malaga. University students, university professors, administration and services staff, and clinical professionals from different health centers participated in this study”.
Also in the methods section, please provide an example of the English and Spanish versions of the questionnaire. Authors: We thank you for your We have submitted The ISAT (English and Spanish version) has been upload as suplementary material.
Also provide an example of what types of questions were used to test internal consistency in both English and Spanish. Authors: We thank you for your Following your suggestion, we have added the questions used to test internal consistency in table 2 (in both English and Spanish).
Finally, can you provide justification on why the COSMIN standards are appropriate for evaluating the internal consistency and criterion validity of the questionnaire. In the citation you provided, the conclusions of the paper pasted below indicates that the COSMIN methods are for systematic reviews and not for the type of work presented in the manuscript."The COSMIN Risk of Bias checklist was developed exclusively for use in systematic reviews of PROMs to distinguish this application from other purposes of assessing the methodological quality of studies on measurement properties, such as guidance for designing or reporting a study on the measurement properties." Authors: We thank you for your The COSMIN is one of the most widespread and comprehensive tool both to assess the risk of bias of primary studies, and measurement properties of instruments for evaluative purposes. A COSMIN Study Design Checklist for Patient-Reported Outcome Measurement Instruments is available online: https://www.cosmin.nl/wp-content/uploads/COSMIN-study-designing-checklist_final.pdf. We have changed the citation as follows:
“Mokkink, L. B.; Terwee, C. B.; Patrick, D. L.; Alonso, J.; Stratford, P. W.; Knol, D. L.; Bouter, L. M.; de Vet, H. C. W. The COSMIN Checklist for Assessing the Methodological Quality of Studies on Measurement Properties of Health Status Measurement Instruments: An International Delphi Study. Qual Life Res 2010, 19 (4), 539–549. https://doi.org/10.1007/s11136-010-9606-8”.
Reviewer 2 Report
Summary: Thank you for the opportunity to review this paper. This paper presents on the reliability and validity of the Spanish translated International Sedentary Assessment Tool (ISAT-Sp). The study included a sample of 432 Spanish speaking adult participants who completed the ISAT-Sp. The authors indicate that the translation and psychometric testing of the ISAT-Sp is important as there is currently no known Spanish self-report measure for collecting sedentary behaviour. Unfortunately, this is false as there have been validations of sedentary measures in Spanish-speaking populations (some are listed below), the authors would do better to say they are scarce. However, given the scarcity in this population this study is warranted. It is also important given the need for the surveillance of this behaviour in population health surveys. I found the paper difficult to follow at times and to parse out the complete methods undertaken. In particular the methods for checking reliability (i.e. were two time points compared) and for validation were not documented. I have several points, both minor and major that I feel need to be addressed in order for the paper to be acceptable for publication. These can be found below.
Abstract: The acronym for sedentary behavior should be changed to “SB” and used consistently throughout the paper. Abstract: In the background it should be noted that the ISAT was developed for use in population health surveys. Throughout the paper please ensure that “the” appears before ISAT. Keywords: suggest adding questionnaire Introduction: The Tremblay et al. paper that you reference for #3 is considered the standard definition of sedentary behaviour. Please remove the statement about “no standard definition of SB”. Introduction, line 37: reclining should be added to the list of postures for SB. Introduction, reference #1 is for physical inactivity which is not the same as sedentary behaviour. Introduction: consider adding reference to the relationship between SB and cardiometabolic health. Introduction, lines 38-39: I’m not sure I understand the meaning behind SB remaining a common problem that is on the rise despite the health benefits associated with regular physical activity. I think the authors mean to say that sedentary behaviour is increasing. Interestingly, however, the international literature examining trends in SB has shown that in general total sedentary time as remained relatively stable in the last 10-20 years, but that time spent using screens especially in leisure time has increased. Introduction: the authors should spend more time describing what the ISAT assesses and its benefits over other measures. Line 51: there should be a space between “weekday” and the references. Lines 51-55: Consider rewriting these sentences to: “The International Sedentary Assessment Tool (ISAT) is an instrument that was recently proposed to assess SB on population health surveys. The ISAT was developed based on the most valid and reliable questions for targeting important modes of SB by systematically reviewing the literature to identify measures with the best psychometric properties. The psychometric properties of the ISAT itself have not been assessed.” Methods: define “adult” i.e. mean age ≥18 years? Methods: were participants community-dwelling? How were they recruited? Please provide more information so that readers can understand the generalizability of the findings. Figure 1: Can you transform this into a study flow diagram? For the last bubble and the figure title please put “ISAT-Sp” to be clear on version. Psychometric testing: What was the ISAT compared to for criterion validity? This is never described in the methods, was it compared to another measure of total sitting? The Discussion says it was not compared to an accelerometer, but criterion validity is described. In the Results, the authors should reiterate between what measures were the correlations corresponding to. Sectoin 2.4: The ISAT was not recommended for use in clinical and research settings, the paper by Prince et al. PeerJ 2017 describes the ISAT being proposed for use in population health surveys. Section 2.4, line 95: What are the scores the authors discuss? The ISAT was developed to get continuous hours and minutes of sedentary time in the various modes. Line 102: This is missing a reference. Line 103: Revise to: “…analyzed using SPSS Version 20 (IBM Corp, NY, USA).” Line 106: In relation to the sample what wasn’t normally distributed? Lines 107-108: Consider rewriting to: “A total of 551 surveys were distributed and 452 were returned. Twenty cases were removed to items missing more than 25%.” What does missing more than 25% refer to? Table 1: The heading for the second column should be mean (SD) or frequency (%). The +/- is not needed with brackets. Change gender to “sex”. Define SD below. Under the variable column, only include the variable and the units. Results: describe the results of the face validity and feedback. As this is a large part of the study, these results should be presented. If they are long they can be included as online supplemental material. Line 118: I don’t see these statistics in Table 2. Line 117: How was overall internal consistency measured? Methods: The authors need to be aware that the ISAT was not developed to be used as a summative measure. The single global sitting item provides an estimate of daily sitting whereas the other questions provide mode-specific information (e.g., time spent sedentary in a travel, sedentary and reading, and sedentary and screen time). Describe what the “items” in Table 2 represent. Table 2: The methods do not describe how the “corrected item-total correlation” is assessed/calculated. For reliability is the measure asked to each participant twice in time? Discussion: The Discussion needs to be strengthened, as it stands now it does not provide much discussion with how the ISAT-Sp compares to other measures, how the global sitting numbers might compare to other studies/surveys which have measured SB in a Spanish population. Did this study take place in Spain? If so, would the results potentially be different than if it had been administered to Spanish speaking Americans? How was it culturally validated? How did the ICCs relate to those reported for the measures within the Prince et al. paper? Strengths and Limitations: be mindful that only the global sitting time question could be validated against an accelerometer, but even then accelerometers measures sedentary time which could also capture stationary standing. Ideally it would be compared to an inclinometer such as an activPAL to compare sitting to sitting. If a log was used then time spent in behaviours using time stamped inclinometer data could be compared for screen and travel time etc. Line 161: What is structural validity? The authors may consider discussing a recent paper by Prince et al. (https://link.springer.com/article/10.1186/s13690-019-0380-y) which also discusses the development of the ISAT.
The authors may wish to consult the following papers which describe SB in Spanish populations or the validation of measures in Spanish populations:
https://journals.plos.org/plosone/article?id=10.1371/journal.pone.0217362
https://bmcpublichealth.biomedcentral.com/articles/10.1186/1471-2458-14-972
https://www.atherosclerosis-journal.com/article/S0021-9150(15)30151-9/fulltext
https://www.nature.com/articles/0801434
Author Response
ITEMIZED LIST OF THE REVIEWER COMMENT
International Journal of Environmental Research and Public Health
Title: Cross-cultural adaptation and psychometric testing of theInternational Sedentary Assessment Tool for the Spanish population.
Version: 2
Date: 07thJanuary 2020
Authors: Dr. Antonio Cuesta-Vargas *, Cristina Roldan-Jimenez, JaimeMartin-Martin, Manuel Gonzalez-Sanchez, Daniel Gutiérrez-Sánchez.
Dear Editors,
We would like to thank you and the reviewers for the review of our manuscript, “Cross-cultural adaptation and psychometric testing of the International Sedentary Assessment Tool for the Spanish population” and the opportunity to revise it. As requested, below we provide a point-by-point response to all reviewer comments and questions. We have added a revised manuscript as well as a copy in which the changes are indicated in red. We believe these revisions have improved the manuscript and look forward to your decision. We are happy to make additional revisions if necessary.
Reviewer 2
Summary: Thank you for the opportunity to review this paper. This paper presents on the reliability and validity of the Spanish translated International Sedentary Assessment Tool (ISAT-Sp). The study included a sample of 432 Spanish speaking adult participants who completed the ISAT-Sp. The authors indicate that the translation and psychometric testing of the ISAT-Sp is important as there is currently no known Spanish self-report measure for collecting sedentary behaviour. Unfortunately, this is false as there have been validations of sedentary measures in Spanish-speaking populations (some are listed below), the authors would do better to say they are scarce. However, given the scarcity in this population this study is warranted. It is also important given the need for the surveillance of this behaviour in population health surveys. I found the paper difficult to follow at times and to parse out the complete methods undertaken. In particular the methods for checking reliability (i.e. were two time points compared) and for validation were not documented. I have several points, both minor and major that I feel need to be addressed in order for the paper to be acceptable for publication. These can be found below. Authors: We really appreciate your comment. According to your comment we have removed the statement “To the best of our knowledge, the Spanish version of the ISAT is the first instrument for assessing SB in Spanish people”.
Abstract: The acronym for sedentary behavior should be changed to “SB” and used consistently throughout the paper. Authors: We really appreciate your comment. According to your comment, we have stated “SB” instead of “SD”.
Abstract: In the background it should be noted that the ISAT was developed for use in population health surveys. Authors: Thank you very much for your comment. Following your suggestion, we have added a new statement as follows: “In this context, the International Sedentary Assessment Tool (ISAT) is an instrument that has been recently developed to assess SB on population health surveys [15]”.
Throughout the paper please ensure that “the” appears before ISAT. Authors: Thank you very much for your comment. According to your comment, we have stated “the” before “ISAT”.
Keywords: suggest adding questionnaire Authors: Thank you very much for your suggestion. Following it, we have added “Questionnaire”. Introduction: The Tremblay et al. paper that you reference for #3 is considered the standard definition of sedentary behaviour. Please remove the statement about “no standard definition of SB”. Authors: Thank you very much for your suggestion. According to your comment, we have removed the statement about “no standard definition of SB”.
Introduction, line 37: reclining should be added to the list of postures for SB. Authors: Thank you very much for your suggestion. We have added “reclining” to the list of postures for SB.
Introduction, reference #1 is for physical inactivity which is not the same as sedentary behaviour. Authors: Thank you very much for your suggestion. We have removed this reference.
Introduction: consider adding reference to the relationship between SB and cardiometabolic health. Authors: Thank you very much for your suggestion. We have added the following reference: “Chastin, S. F. M.; Egerton, T.; Leask, C.; Stamatakis, E. Meta-Analysis of the Relationship between Breaks in Sedentary Behavior and Cardiometabolic Health. Obesity 2015, 23 (9), 1800–1810. https://doi.org/10.1002/oby.21180”.
Introduction, lines 38-39: I’m not sure I understand the meaning behind SB remaining a common problem that is on the rise despite the health benefits associated with regular physical activity. I think the authors mean to say that sedentary behaviour is increasing. Interestingly, however, the international literature examining trends in SB has shown that in general total sedentary time as remained relatively stable in the last 10-20 years, but that time spent using screens especially in leisure time has increased. Authors: Thank you very much for your comment. We mean to say that despite there is evidence for the health effects associated with regular physical exercise, SB remains a common problem. We have deleted the statement “that is on the rise”.
Introduction: the authors should spend more time describing what the ISAT assesses and its benefits over other measures. Authors: Thank you very much for your comment. We have rephrased the paragraph where the ISAT has been explained. We have introduced the following paragraph to describe what ISAT assesses and its benefits over other measures: “The ISAT is a questionnaire aimed at evaluating SB and that has been composed from the selection of the most valid and reliable questions related to specific aspects of the SB identified after a systematic review in which 49 questionnaires oriented to the evaluation of the SB were included both in adults and in pediatric patients”.
Line 51: there should be a space between “weekday” and the references. Authors: Thank you very much for your comment. According to your comment, we have introduced a space between “weekday” and the references.
Lines 51-55: Consider rewriting these sentences to: “The International Sedentary Assessment Tool (ISAT) is an instrument that was recently proposed to assess SB on population health surveys. The ISAT was developed based on the most valid and reliable questions for targeting important modes of SB by systematically reviewing the literature to identify measures with the best psychometric properties. The psychometric properties of the ISAT itself have not been assessed.” Authors: We really appreciate your comment. Following your recommendation, we have reworded these sentences as follows: “In this context, the International Sedentary Assessment Tool (ISAT) is an instrument that has been recently proposed to assess SB on population health surveys[15]. The ISAT is a questionnaire formed by the most valid and reliable questions related to specific aspects of the SB identified after a systematic review that included 49 questionnaires aimed to evaluate the SB in both adults (35 questionnaires) and pediatric (14 questionnaires) patients [15]. However, the psychometric properties of the ISAT itself have not been assessed”.
Methods: define “adult” i.e. mean age ≥18 years? Authors: We really appreciate your comment. Following your suggestion, we have reworded the statement as follows: “The inclusion criteria were: (1) Spanish-speaking adults (age ≥18 years); (2) people who had signed an informed consent”.
Methods: were participants community-dwelling? How were they recruited? Please provide more information so that readers can understand the generalizability of the findings. Authors: We really appreciate your comment. Different sources were used for the recruitment of participants. These have been explained in the following paragraph introduced in the document: “The participants were recruited from different social environments in Malaga (Spain). University students, university professors, administration and services staff and clinical professionals from different health centers participated in this study”.
Figure 1: Can you transform this into a study flow diagram? For the last bubble and the figure title please put “ISAT-Sp” to be clear on version.
Authors: We really appreciate your comment. Following your suggestion, we have added a flow diagram and ISAT-Sp in the last bubble and the figure title.
Psychometric testing: What was the ISAT compared to for criterion validity? This is never described in the methods, was it compared to another measure of total sitting?
Authors: We used the time that participants have spent sitting on a weekday during the last seven days for criterion validity. This is described in methods (section 2.3. Psychometric Testing) as follows: “The ISAT will be positively correlated with the time that participants have spent sitting on a weekday during the last seven days, with a minimum value of correlation coefficient of 0.50”. In this context, the following paper indicate that total sedentary time is a conventional and current indicator of sedentary behavior:
Yamamoto, K.; Matsuda, F.; Matsukawa, T.; Yamamoto, N.; Ishii, K.; Kurihara, T.; Yamada, S.; Matsuki, T.; Kamijima, M.; Ebara, T. Identifying Characteristics of Indicators of Sedentary Behavior Using Objective Measurements. J Occup Health 2019. https://doi.org/10.1002/1348-9585.12089.
The Discussion says it was not compared to an accelerometer, but criterion validity is described. In the Results, the authors should reiterate between what measures were the correlations corresponding to. Authors: We used the time that participants have spent sitting on a weekday during the last seven days for criterion validity. This is described in methods (section 3. Psychometric Testing) as follows: “The ISAT will be positively correlated with the time that participants have spent sitting on a weekday during the last seven days, with a minimum value of correlation coefficient of 0.50”. Moreover, we reiterated between what measures were the correlations in Results ( Section3.3.2. Criterion validity) as follows: “Concurrent criterion-related validity was determined from the relationship between the ISAT and the time that participants have spent sitting on a weekday during the last seven days. Concurrent criterion-related validity was demonstrated, with a fair and positive correlation of rho= 0.63”.
Section 2.4: The ISAT was not recommended for use in clinical and research settings, the paper by Prince et al. PeerJ 2017 describes the ISAT being proposed for use in population health surveys. Authors: We have reworded the statement as follows: “The ISAT is an instrument that has been developed to assess SB [15]. This instrument has been recommended for use in population health surveys based on the best available evidence”.
Section 2.4, line 95: What are the scores the authors discuss? The ISAT was developed to get continuous hours and minutes of sedentary time in the various modes. Authors: Thank you for your comment. Although the ISAT was developed to get continuous hours and minutes of sedentary time in the various modes, a total score can be obtained. In this regard, higher scores involve more SB.
Line 102: This is missing a reference. Authors: Thank you very much for your correction. We have added the following rererence: “Portney LG, Watkins MP. Foundations of Clinical Research: Applications to Practice. 3rd ed. New York, NY: Pearson/Prentice Hall; 2009.”
Line 103: Revise to: “…analyzed using SPSS Version 20 (IBM Corp, NY, USA).” Authors: Thank you very much for your correction. We have changed this statement as follows: “Data was entered into a data file and analysed using an SPSS statistical program (version 20)”.
Line 106: In relation to the sample what wasn’t normally distributed? Authors: Thank you very much for your question. The total score of ISAT which was used to assess criterion validity wasn’t normally distributed. In this context, The KS test [Asymp. Sig. (2-tailed) = 0.004] indicated that the sample was not normally distributed for this variable.
Lines 107-108: Consider rewriting to: “A total of 551 surveys were distributed and 452 were returned. Twenty cases were removed to items missing more than 25%.” What does missing more than 25% refer to? Authors: Thank you very much for your question. Following your suggestion, we have rewriting this statement as follows: “A total of 552 surveys were distributed and 452 surveys were returned. Twenty cases were removed due to items missing (more than 25%)”.
Table 1: The heading for the second column should be mean (SD) or frequency (%). The +/- is not needed with brackets. Change gender to “sex”. Define SD below. Under the variable column, only include the variable and the units. Authors: Thank you very much for your suggestion. According to the suggestion, we have changed the heading, and we have stated “Mean (SD) or frequency (%)”. We have also removed the +/-. Moreover, we have stated “sex” instead of “gender”. Finally, SD has been defined below.
Results: describe the results of the face validity and feedback. As this is a large part of the study, these results should be presented. If they are long they can be included as online supplemental material. Authors; thanks for the suggestions, new sentence was added in the Results section, as follow; “Minor discrepancies were discussed and resolved during face validity and feedback”.
Line 118: I don’t see these statistics in Table 2. Line 117: How was overall internal consistency measured? Authors: Thank you very much for your comment. The overall internal consistency was calculated with all ISAT-Sp items. As we stated in Data analysis section, Cronbach’s α coefficients and Intraclass Correlation Coefficient Type 2.1 (ICC2.1) were calculated to determine the internal consistency of ISAT. We have reworded the statement as follows: “The overall internal consistency of the questionnaire was α =0.80, and the intra-class correlation coefficient was 0.80 (95% CI 0.75 to 0.84)”.
Methods: The authors need to be aware that the ISAT was not developed to be used as a summative measure. The single global sitting item provides an estimate of daily sitting whereas the other questions provide mode-specific information (e.g., time spent sedentary in a travel, sedentary and reading, and sedentary and screen time). Describe what the “items” in Table 2 represent. Authors: Thank you very much for your comment. Following the reviewers suggestions, we have added the questions used to test internal consistency in table 2 (in both English and Spanish).
Table 2: The methods do not describe how the “corrected item-total correlation” is assessed/calculated. For reliability is the measure asked to each participant twice in time? Authors: Thank you very much for your comment. Item total correlation was assessed with the Intraclass Correlation Coefficient Type 2.1 (ICC2.1) (Two-way random effects, absolute agreement, single rater/measurement).
For reliability, we have assessed internal consistency. Internal consistency assesses the correlation between multiple items in a test that are intended to measure the same construct. We have calculated internal consistency without repeating the test or involving other researchers, so it’s a good way of assessing reliability when you only have one data set.
Discussion: The Discussion needs to be strengthened, as it stands now it does not provide much discussion with how the ISAT-Sp compares to other measures, how the global sitting numbers might compare to other studies/surveys which have measured SB in a Spanish population.
Did this study take place in Spain? If so, would the results potentially be different than if it had been administered to Spanish speaking Americans? How was it culturally validated?
Authors: We really appreciate your comment. As you requested in a previous comment, we have added a new statement in Methods section as follows: “The participants were recruited from different social environments in Malaga (Spain). University students, university professors, administration and services staff and clinical professionals from different health centers participated in this study”. Moreover, we have added a new statement to strength the Discussion section as follows: “In comparison with other measures which have been validated for assessing SB in Spanish population, our results showed higher values of convergent validity than those found by Munguía-Izquierdo et al. [30]”.
Concerning the results in other Spanish speaking populations, they would potentially be different if ISAT-Sp had been administered to Spanish speaking Americans. In this context, the Spanish is one of the languages questionnaires are most often adapted to, the native speakers being distributed in more than 30 countries and having large cultural differences. Although it is not usually to find adaptations from American English to Australian or British English in English-speaking contexts like in Spanish, these do exist (Vallejo-Medina et al., 2017). Furthermore, SB assessment can be influenced by several elements, such as sociocultural factors from the context in which it is evaluated.
Vallejo-Medina, P.; Gómez-Lugo, M.; Marchal-Bertrand, L.; Saavedra-Roa, A.; Soler, F.; Morales, A. Developing Guidelines for Adapting Questionnaires into the Same Language in Another Culture. Terapia psicológica 2017, 35 (2), 159–172. https://doi.org/10.4067/s0718-48082017000200159.
How did the ICCs relate to those reported for the measures within the Prince et al. paper?
Authors: Thank you very much for your comment. According to your comment, we have added a statement about the ICC and Cronbach’s α, as well as Criterion validity and their relationship to Prince et al. Paper in Discussion section as follows:
“The internal consistency was satisfactory (Cronbach’s α = 0.80), and the intra-class correlation coefficient was 0.80 (95% CI 0.75 to 0.84). According to Prince et al, it is ideal to have an ICC and Cronbach’s α as close to 1 as possible, with values over 0.75 considered excellent, for a measure of sedentary behaviour in population health surveys [15]. In this regard, the values of ICC and Cronbach’s α for the ISAT-Sp were excellent, and were higher than those reported in other studies on the validation of instruments assessing SB [15,17, 26-27].
Criterion validity analysis was supported by a fair correlation (rho= 0.63), which provides evidence of construct validity. Validity of a self-report SB measures have been often assessed against a objective measure such as an accelerometer [15]. Validation studies of self-report SB measures have shown poor criterion validity when these measures have been assessed against objective measures [15]. In this context, the values of criterion validity for the ISAT-Sp were high when compared to other studies on the validation of instruments assessing SB [15,28,29]”.
Strengths and Limitations: be mindful that only the global sitting time question could be validated against an accelerometer, but even then accelerometers measures sedentary time which could also capture stationary standing. Ideally it would be compared to an inclinometer such as an activPAL to compare sitting to sitting. If a log was used then time spent in behaviours using time stamped inclinometer data could be compared for screen and travel time etc. Authors: Thank you very much for your comment. The absence of a device for making objective records during the time that the patient remains immobile has been introduced as part of the limit. In addition, the importance of using, as the reviewer indicates, inclinometers, capable of differentiating between standing and sitting positions has been emphasized.
Line 161: What is structural validity? Authors: Thank you very much for your comment. According to COSMIN standards, structural validity is “the degree to which the scores of an instrument are an adequate reflection of the dimensionality of the construct to be measured”. It refers to the model fit of a factor analysis on all items in an outcome measurement instrument, e.g. to confirm a 3‐factor model for an instrument with three subscales.
The authors may consider discussing a recent paper by Prince et al. (https://link.springer.com/article/10.1186/s13690-019-0380-y) which also discusses the development of the ISAT. Authors: Following your suggestion, we have discussed this paper. We have added a new statement in discussion section and a new reference as follows:
“There is a need for the development of valid and reliable instruments for measuring SB to provide accurate and consistent measures over time [31]. These instruments must be concise, valid and reliable, evidence-based, and developed using best practices [31]. In this context, the ISAT is a measure that was recently proposed to assess SB on population health surveys [15]. The ISAT was developed based on the most valid and reliable questions for targeting important modes of SB by systematically reviewing the literature to identify measures with the best psychometric properties. This study provides evidence for the validity of the ISAT-Sp. In this regard, the ISAT-Sp is an instrument concise, valid and reliable, evidence-based, and developed using best practices. Thus, the ISAT-Sp is a measure that can be recommended to assess SB on population health surveys in Spanish population”.
The authors may wish to consult the following papers which describe SB in Spanish populations or the validation of measures in Spanish populations: https://journals.plos.org/plosone/article?id=10.1371/journal.pone.0217362 https://bmcpublichealth.biomedcentral.com/articles/10.1186/1471-2458-14-972 https://www.atherosclerosis-journal.com/article/S0021-9150(15)30151-9/fulltext https://www.nature.com/articles/0801434 https://pdfs.semanticscholar.org/18aa/3558d6a44ca024d56a880891058329aa52ff.pdf
Authors: We really appreciate your comment. Following your recommendation, we have consulted these papers and we have added a new statement about the validation of a measure for assessing SB in Spanish population. Moreover, we have added a new reference: Munguia-Izquierdo, D.; Segura-Jiménez, V.; Camiletti-Moirón, D.; Alvarez-Gallardo, I. C.; Estévez-López, F.; Romero, A.; Chillon, P.; Carbonell-Baeza, A.; Ortega, F. B.; Ruiz, J. R.; et al. Spanish Adaptation and Psychometric Properties of the Sedentary Behaviour Questionnaire for Fibromyalgia Patients: The al-Andalus Study. Clin. Exp. Rheumatol. 2013, 31 (6 Suppl 79), S22-33
Round 2
Reviewer 1 Report
The authors have properly addressed all my comments.